# Mammalian Brain Ca^2+^ Channel Activity Transplanted into *Xenopus laevis* Oocytes

**DOI:** 10.3390/membranes12050496

**Published:** 2022-05-02

**Authors:** Matthieu Rousset, Sandrine Humez, Cyril Laurent, Luc Buée, David Blum, Thierry Cens, Michel Vignes, Pierre Charnet

**Affiliations:** 1IBMM, UMR 5247 CNRS, Université de Montpellier, ENSCM, 1919 Route de Mende, 34293 Montpellier, France; thierry.cens@inserm.fr (T.C.); michel.vignes@umontpellier.fr (M.V.); 2Lille Neuroscience & Cognition, Université de Lille, F-59000 Lille, France; sandrine.humez@univ-lille1.fr (S.H.); cyril.laurent@univ-lille1.fr (C.L.); luc.buee@inserm.fr (L.B.); david.blum@inserm.fr (D.B.); 3Inserm UMR_S1172, Jean-Pierre Aubert Research Centre, F-59000 Lille, France; 4Lille Neuroscience & Cognition, Alzheimer & Tauopathies, CHU-Lille, F-59000 Lille, France

**Keywords:** membrane microtransplantation, voltage clamp, Ca_V_2 Ca^2+^ channels, channelopathies

## Abstract

Several mutations on neuronal voltage-gated Ca^2+^ channels (VGCC) have been shown to cause neurological disorders and contribute to the initiation of epileptic seizures, migraines, or cerebellar degeneration. Analysis of the functional consequences of these mutations mainly uses heterologously expressed mutated channels or transgenic mice which mimic these pathologies, since direct electrophysiological approaches on brain samples are not easily feasible. We demonstrate that mammalian voltage-gated Ca^2+^ channels from membrane preparation can be microtransplanted into *Xenopus* oocytes and can conserve their activity. This method, originally described to study the alteration of GABA receptors in human brain samples, allows the recording of the activity of membrane receptors and channels with their native post-translational processing, membrane environment, and regulatory subunits. The use of hippocampal, cerebellar, or cardiac membrane preparation displayed different efficacy for transplanted Ca^2+^ channel activity. This technique, now extended to the recording of Ca^2+^ channel activity, may therefore be useful in order to analyze the calcium signature of membrane preparations from unfixed human brain samples or normal and transgenic mice.

## 1. Introduction

The microtransplantation technique makes a strong a priori that any channel or receptor that is functionally active in its neuronal native membrane will conserve its functionality throughout the entire membrane preparation procedure and oocyte injection, and will be properly inserted (with its surrounding native membrane) into the *Xenopus* oocyte membrane [1,2,3,4,5]. It is therefore a very useful technique used to study membrane receptors or channels in their native environment, which are not technically or ethically accessible through usual electrophysiological techniques (whole-cell recording on isolated neurones or slices), such as human samples [5]. Although this technique has been successfully used to study GABA-A or nicotinic receptors from torpedo, rat, or human tissues [1,2,3,4,5,6,7], its application to analyze voltage-gated Ca^2+^ channels has only been attempted using indirect measurement of channel activity [8]. This extension is not obvious, since these channels are subject to a relatively strong current run-down during the course of whole-cell recording experiments reportedly due to the wash-out of essential intracellular components [9,10]. Indeed, membrane-localized components such as PIP2 [11], as well as intracellular regulatory proteins such as kinases [9], have been shown to contribute to this phenomenon, and thus questioned the possibility of keeping normal Ca^2+^ channel activity during the course of membrane preparation and oocyte injection. However, the fact that a number of kinase activities are present in the Xenopus oocyte, as well as the recording of single-channel activity in artificial or reconstituted membrane bilayers, are in favor of a successful functional transplantation of channel activity.

In this work, we show the functional reconstitution of voltage-gated Ca^2+^ channel activity in Xenopus oocytes that have been injected with hippocampal, cerebellum, or cardiac membrane preparations. The channel activity remained high and was regulated by neuronal G-protein-coupled receptors when co-expressed together with the transplanted channels.

## 2. Materials and Methods

All animals were maintained in standard animal cages under conventional laboratory conditions (12 h/12 h light/dark cycle, 22 °C), with ad libitum access to food and water. The animals were maintained in compliance with European standards for the care and use of laboratory animals and experimental protocols approved by the local Animal Ethical Committee (No. CEEA342012 on 12 December 2012).

### 2.1. Membrane Preparation

Mice were killed by cervical dislocation. The whole brain was rapidly removed from the skull and immersed for 1 min in ice-cold artificial cerebrospinal fluid (ACSF) solution containing (in mM): NaCl 117, KCl 4.7, CaCl_2_ 2.5, MgCl_2_ 1.2, NaH_2_PO_4_ 1.2, NaHCO_3_ 25, and glucose 10. The ACSF was continuously oxygenated with 95% O_2_ and 5% CO_2_ to maintain the proper pH (7.4). Brain areas were dissected on dampened Whatman paper with ACSF on an ice-cold petri dish. After separation of the two hemispheres, achieved by running a scalpel blade through the intrahemispheric fissure, the posterior part of the brain (midbrain, pons, medulla, and cerebellum) was removed from each hemisphere. After placing the medial face of a hemisphere facing up, tissues recovering the medial surface of the hippocampus (thalamus septum and striatum) were gently pulled up using one spatula after inserting its tip right through the corpus callosum, while the other tissues were maintained with another spatula. A spatula was then inserted under the ventral part of the hippocampus and used to roll the hippocampus and to separate it from the cortex.

Membranes were prepared as described [6], with slight modifications for hippocampus, cerebellar, and cardiac ventricle preparation from 8–12-week-old C57BL/6 male mice (Charles River, City, France). Using a Teflon glass homogenizer, about 0.5 g of tissue was homogenized in 400 µL ml of glycine buffer (in mM): glycine (200), NaCl (150), sucrose (300), EGTA (50), and EDTA (50), as well as 20 µL of protease inhibitors (Sigma P2714), pH9 with NaOH. The filtrate was centrifuged for 15 min at 9500× *g* in a Beckmann centrifuge (C1015 rotor). The supernatant was then centrifuged for 2 h at 100,000 *g* in an SW40 rotor at 4 °C. The pellet was washed, resuspended in 30 µL of 5 mM glycine, and used directly for the evaluation of protein concentration or aliquoted and kept at −80 °C for later use. This membrane preparation was diluted at 5 mg/mL for injection into freshly isolated Xenopus oocytes. 

### 2.2. Xenopus Oocytes Isolation and Injection

*Xenopus* oocytes were isolated from anesthetized female *Xenopus laevis*, as already described [12], and kept at 18 °C in ND96 solution (in mM): NaCl (96), KCl (2), CaCl_2_ (1.8), MgCl_2_ (2), HEPES (5, pH = 7.2 with NaOH), and under gentle agitation. The day following the oocytes’ preparation, 50–100 nl of membrane preparation (0.2–10 mg protein/mL in 5mM glycin) was injected using injection needles pulled from capillary tubes (Harward Apparatus, GC150T10) on a vertical puller (Sutter Instrument, P-30). Macroscopic currents were recorded under two electrode voltage clamps 5–96 h later. 

### 2.3. Calcium Current Recordings

GABA receptor activity was recorded in ND96 supplemented with 1 mM GABA (γ-aminobutyric acid). The solution for recording Ca^2+^ channel activity (Bant40) had the following compositions (in mM): Bant40: BaOH_2_ (40), TEAOH (20), NMDG (20), CsOH (2), and HEPES (10, pH = 7.2 with methanesulfonic acid). Currents were filtered (500 Hz) and digitized (2 kHz) using a Digidata-1200 interface (Axon Instruments). Data acquisition was performed using version 9 of the pClamp software (Axon Instruments). Around 30 nl of BAPTA (in mM: BAPTA free acid, 100; CsOH, 10; HEPES, 10; pH 7.2 CsOH) was injected into each oocyte (10 psi, 150 msec) at the beginning of each Ca^2+^ channel recording using a third electrode to minimize contamination by the Ca^2+^-activated Cl^−^ current. Isochronal steady-state inactivation curves (2.5 s of conditioning voltage followed by a 400 ms test pulse to +10 mV) were fitted using the following equations.

I/Imax = R_in_+(1 − R_in_)/(1 + exp((V − V_in_)/k_in_)) where I: the current amplitude measured during the test pulse to +10 mV after a conditioning voltage steps varying from −80 to +50 mV; Imax: the current amplitude measured during the test pulse for a conditioning step to −80 mV; V_in_: the potential for half-inactivation; V: the conditioning voltage; k_in_: the slope factor; and R_in_: the proportion of non-inactivating current. Current to voltage curves were fitted using the following equation:

I/Imax= G*(V − E_rev_)/(1 + exp((V − V_act_)/k_act_)) where I: the current amplitude measured during depolarization varying from −80 to +50 mV; Imax: the peak current amplitude measured at the maximum of the current–voltage curve; G: the normalized macroscopic conductance; E_rev_: the apparent extrapolated reversal potential; V_act_: the potential for half-activation; V: the value of depolarization; and k_act_: the slope factor.

### 2.4. Data Analysis

Results were expressed as the mean ± *SEM*. The statistical significance of differences between two groups was determined using the non-paired Student’s *t*-test. 

## 3. Results

### 3.1. Characterization of Membrane-Transplanted Oocytes 

The effects of microtransplantation/injection of 50 nl of adult hippocampal mouse membrane preparation (at 5 mg/mL) into Xenopus oocytes were first analyzed on two basic properties of Xenopus oocytes that could be potentially affected: the membrane potential and the oocytes’ capacitance (Appendix A left and right). The oocytes’ resting potential was measured in three batches of oocytes: non-injected and membrane-injected oocytes, one or two days post-injection. This potential was not significantly different between the different batches with average values of around −35 mV (−36 ± 2 mV; −40 ± 3 mV and −36 ± 42 mV; for resting potentials of control and one or two days after membrane-injection into oocytes, respectively), similar to what has been previously reported for control, non-injected, or water or 5 mM glycine-injected oocytes, although these values can vary substantially (from −25 mV to −50 mV [13]). It is interesting to note, however, that while the resting potential between day 1 and day 2 was not significantly different in all the batches of membranes tested, it was nevertheless slightly more negative at day 1 than at day 2 (−40 mV and −36 mV for this batch for example). Similarly, the oocyte membrane capacitance was not modified by the injection of membrane preparation and remained similar to the value of control-injected oocytes (231 ± 4 nF, *n* = 11 and 228 ± 5 nF, *n* = 11, respectively [13]). Therefore, it appears that the injection of plasma membrane into oocytes did not significantly modify the cell capacitance (and thus in the first approximation, the membrane surface) and the membrane potential, suggesting that the activity of the inserted ion channels and the quantity of the inserted membrane were modest and/or partly balanced via endogenous regulation and/or membrane recycling.

We then tested the response of these oocytes to 1 mM GABA. As expected, non-injected oocytes (Figure 1A, N.I.), water-injected oocytes (not shown), or 5 mM glycine-injected oocytes (not shown) did not respond to the perfusion of GABA, while in oocytes injected a few hours before with hippocampal membrane preparation, a clear inward current that could reach several hundreds of nA was recorded. This current was carried by Cl^−^ ions, displayed an EC_50_ for GABA of 128 ± 31 µM (*n* = 4), and was blocked by 87 ± 2% (*n* = 5) by 100 µM biccuculine (not shown). The recording of native GABA receptors started a few hours after injection (>5 h, D0, Figure 1B), and reached its maximum the day after (D1) with no further increase at 48 h (D2, Figure 1B). The expression of GABA receptors was also not modified when the concentration of the membrane preparation (estimated by a Bradford assay) was double (7 in place of 3.5 mg/mL), suggesting that the process of incorporation of mouse membrane into the oocyte plasma membrane was already saturated. Further experiments were thus performed 24 h post-injection (D1), and using a membrane preparation at 5 mg/mL.

### 3.2. Expression of Voltage-Gated Ca^2+^ Channels in Transplanted Oocytes

We then tested the Ca^2+^ channel activity in these oocytes injected with hippocampal membrane and bathed with the appropriate solution (no Na^+^, no Cl^−^, and 40 mM Ba^2+^, see methods). Usually, an endogenous Ba^2+^ current that never exceeded 50 nA in 40 mM Ba^2+^ (and could not be detected in 10 mM Ba^2+^) was recorded in non-injected oocytes ([14,15], Figure 2D). This current was even lower when the endogenous Ca^2+^-activated Cl^−^ current was blocked by injection of BAPTA into the oocyte [16]. In the membrane-injected oocytes, the Ba^2+^ current reached more than 100 nA a few hours after injection (Figure 2A) and could be larger than 500 nA the day after, without further increase at D2 (Figure 2B), as already seen in the case of the GABA-induced current. The average Ba^2+^ current recorded variations from injection to injection but was, on average, in the 200–400 nA range for hippocampal preparation, with a maximum of 700 nA, close to what was obtained following mRNA injection from brain preparation [17,18]. Since this expression occurred without any marked increase in the Xenopus oocyte membrane, we can suppose that the amount of incorporated brain membrane was less than 1% of the total oocyte plasma membrane, which thus represented less than 2 nF, giving a current RNA of 200–400 nA/2nF, i.e., 100–200 pA/pF. Although these calculations are subject to many unknown factors, such as the level of oocyte membrane recycling, they nevertheless suggest that the channel activity transplanted is close to the normal brain channel activity with values of 10–100 pA/pF (cerebellar Purkinje neurons [19,20] and granule cells [21], or hippocampal neurons [22,23], for example). Interestingly, besides these variations, when the ratio of the expressed Ba^2+^ current was normalized to the GABA current (I_Ba_/I_GABA_) recorded using different control membrane preparations, but in the oocytes of the same batch, very little variability was recorded (Figure 2C), demonstrating that at least these two channels can be recovered with a good reproducibility, and that this ratio can be used to analyze any variation in the relative expression of these channels in diseased or mutant animals. 

Injections of membrane preparations of different origins (cerebellum or cardiac ventricle) produced Ba^2+^ currents with different average amplitudes. The hippocampal membrane was clearly more efficient than the cerebellar membrane (Figure 2D), with Ba^2+^ currents displaying a 2–3 times higher average amplitude (3 preparations tested). Cardiac membrane preparations were also less efficient, with Ba^2+^ current significantly higher than the control current recorded in non-injected oocytes, but 2–3 times smaller than those from “hippocampal” oocytes. Different membrane preparations of these tissues were tested with similar results, suggesting that either the cerebellar and cardiac channels were less efficiently isolated within the membrane preparation, our membrane preparation protocol used for hippocampal membrane was inappropriate for cerebellar/cardiac tissues, and/or these preparations were not able to induce functionally active channels after membrane injection due to a lack of associated proteins or signalling pathways. Experiments are underway to try and solve these issues.

### 3.3. Biophysical and Pharmacological Properties of Expressed VGCC

Analysis of the current–voltage and isochronal inactivation curves in oocytes injected with hippocampal membrane preparation gave potentials for half activation and half inactivation of −5 ± 1 mV and −25 ± 3 mV, respectively, close to the values obtained from brain mRNA-injected oocytes with a threshold for activation of −40 mV and a peak of the current–voltage curves at 10–20 mV in 40 mM Ba^2+^ [24]. These values were similar for oocytes transplanted with cerebellar membrane, with a slight hyperpolarizing shift (−6 mV) in the inactivation curve (Figure 3A–C). Although we did not perform an extensive pharmacological analysis of these channels produced by membrane transplantation, the sensitivity to nifedipine, a specific L-type Ca^2+^ channel antagonist, was tested on each membrane preparation (hippocampal, Figure 4A), but always without success, strongly suggesting that the only channel activity that could be transplanted was produced by a non L-type channel subunit. Moreover, since we could also never record any low threshold T-type Ca^2+^ channel, we concluded that Ca^2+^ channels encoded by the Ca_V_1 and Ca_V_3 genes could not be transplanted using hippocampal membrane preparation. 

Channels encoded by the Ca_V_2 family are tightly regulated by a G-protein-dependent regulation that is activated upon stimulation of the G-protein-coupled 7 transmembrane domain receptors [25]. In sensitive neurons, for example, the µ-opioid receptor µOR has been shown to down-regulate Ca^2+^ channel activity in a voltage-dependent manner [26,27]. In hippocampal neurons, however, this regulation has never been described, and DAMGO, a specific µ-opioid agonist, has been reported to have no effect on hippocampal Ca^2+^ currents [28]. However, the hippocampal Ca^2+^ channels are under the influence of the GABA-B receptors and the perfusion of baclofen induced a strong inhibition of channel activity. To test this regulation in our hippocampal membrane-injected oocytes, we tested the effects of the µ-opioid agonist DAMGO (10 µM) on the expressed Ba^2+^ current. The Ba^2+^ current was recorded using voltage steps up to +10 mV, and DAMGO was perfused in the recording solution for several minutes without any marked effects, consistent with the lack of regulation in native neurons (Figure 4B,C). Interestingly, when the same experiment was performed on oocytes injected with the membrane preparation and co-injected with in vitro-synthesized RNA coding for the µOR1 receptor, a clear inhibition of the current, by 18 ± 1% on average, occurred during the perfusion of DAMGO. Moreover, when a two-pulse protocol was applied on these oocytes, where the second pulse was preceded by a strong depolarization to +100 mV, a marked facilitation of the second pulse was noted (+16 ± 12%) in the presence of DAMGO that was absent without DAMGO, which was indicative of the voltage-dependent inhibition specific to the G-protein (not shown). This demonstrated that while the full signaling pathway linking the µ-opioid receptor to the Ca^2+^ channels was not present in the microtransplanted oocytes, it could nevertheless be reproduced by co-expression of the receptor, building a functional link between the channel and the 7 TM receptor. This possibility opened new avenues to analyze the impact of various tissue-specific post-translational modifications on channel regulation independent of the presence of a given receptor in the tissue of origin.

Finally, we checked if the cytoplasmic Ca^2+^ channel Ca_V_β subunit remained associated with the channels during the membrane preparation procedure. Western blots of both hippocampal and cerebellar membrane preparation were treated with a rabbit antibody against all known mammalian Ca_V_β subunits (antiCa_V_β-com, Alomone laboratories) and revealed an anti-rabbit HRP-conjugated antibody. As seen in Figure 4D, several bands were revealed on the membrane at molecular weights close to those expected for neuronal Ca_V_β subunits (see Appendix A), strongly suggesting that at least some cytoplasmic regulatory Ca_V_β subunits remained present in membrane preparation and regulated channel activity.

## 4. Conclusions

Within a few hours, the microtransplantation/injection of adult hippocampal mouse membranes into Xenopus oocytes leads to the appearance of Ca^2+^ current activity (100–700 nA) with properties and regulation similar to neuronal HVA Ca_V_2 Ca^2+^ channels, and with the preservation of their cytoplasmic accessory Ca_V_β subunits. 

The time course of current production was comparable to the appearance of the GABA-induced current, and the relative amplitude of the two types of currents (GABA and Ba^2+^ currents) was quite stable between different membrane preparations of the same tissue. 

This type of membrane preparation was less efficient to record Ca^2+^ channel activity when cerebellar or cardiac tissues were used.

## Figures and Tables

**Figure 1 membranes-12-00496-f001:**
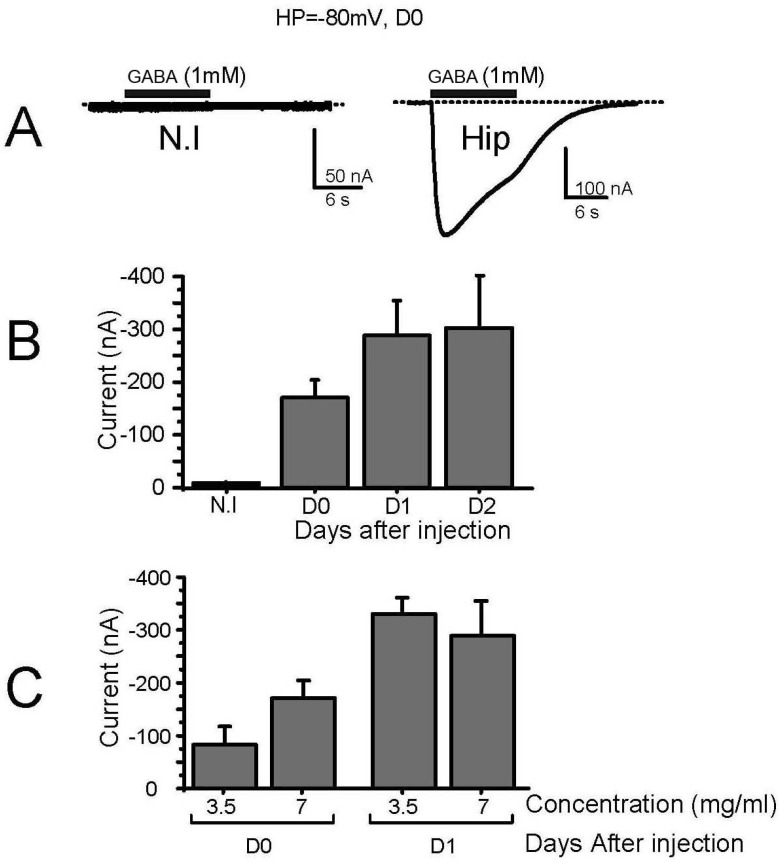
Kinetic of receptor expression after injection of membrane preparation. (**A**) Typical current traces recorded in response to perfusion of GABA (1 mM) of control oocytes (N.I.) or oocytes injected with hippocampal brain membrane preparation (Hip), 6 h after injection (D0). (**B**) Kinetics of the response to GABA of oocytes injected with hippocampal brain membrane preparation quantified as the peak current amplitude recorded in response to GABA application (1 mM, HP = −80 mV, EC50 = 81µM) at D0: 6 h after membrane injection, and D1 and D2: 24 and 48 h after membrane injection, respectively. (**C**) Effect of injected hippocampal membrane concentration on the amplitude of the response of the oocytes to the perfusion of GABA. Two concentrations were used (3.5 and 7 mg/mL) and two time points were analyzed (D0 and D1).

**Figure 2 membranes-12-00496-f002:**
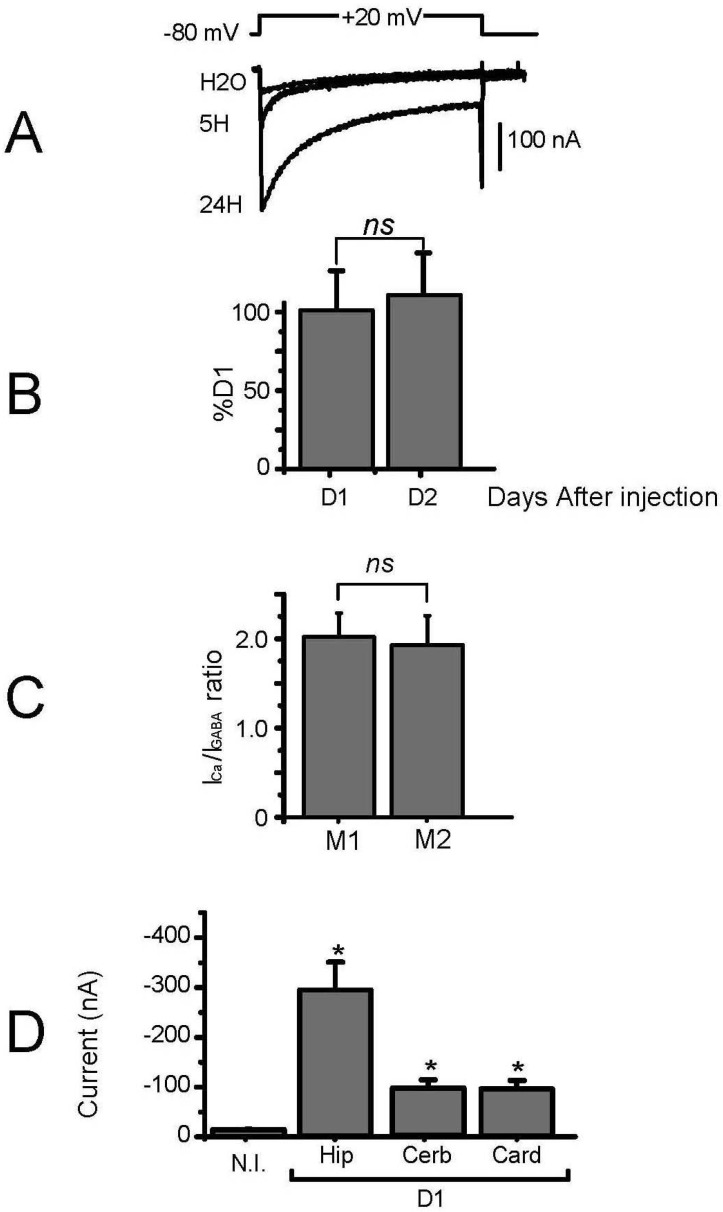
Expression of voltage-gated Ca^2+^ channels after injection of membrane preparation. (**A**) Typical current traces recorded on oocytes injected with water (H_2_O), or hippocampal membrane preparation in response to depolarizations to +20 mV from a holding potential of −80 mV, 5 h (5 H) or 24 h (24 H) after injection. The recording medium contained 40 mM Ba^2+^. (**B**) Expression was at its maximum level one day after injection (D1, set at 100%), and the current amplitude was similar at day 2 (D2) following membrane injection. (**C**) Histogram showing the ratio of the level of Ca^2+^ channel expression (measured as the peak current amplitude recorded during a depolarizing pulse to +10 mV in BANT40) on GABA receptor expression (measured as the peak current amplitude in response to 1 mM GABA at HP of −80 mV in ND86) for two different mice, M1 and M2. (**D**) Different membrane preparations (at 5 mg/mL) produced different levels of voltage-gated Ca^2+^ channel expression (N.I: non-injected oocytes, or oocytes injected with Hip: hippocampal membrane, Cerb: cerebellum membrane or Card: cardiac ventricular membrane). ns means—not significant, *—with injection.

**Figure 3 membranes-12-00496-f003:**
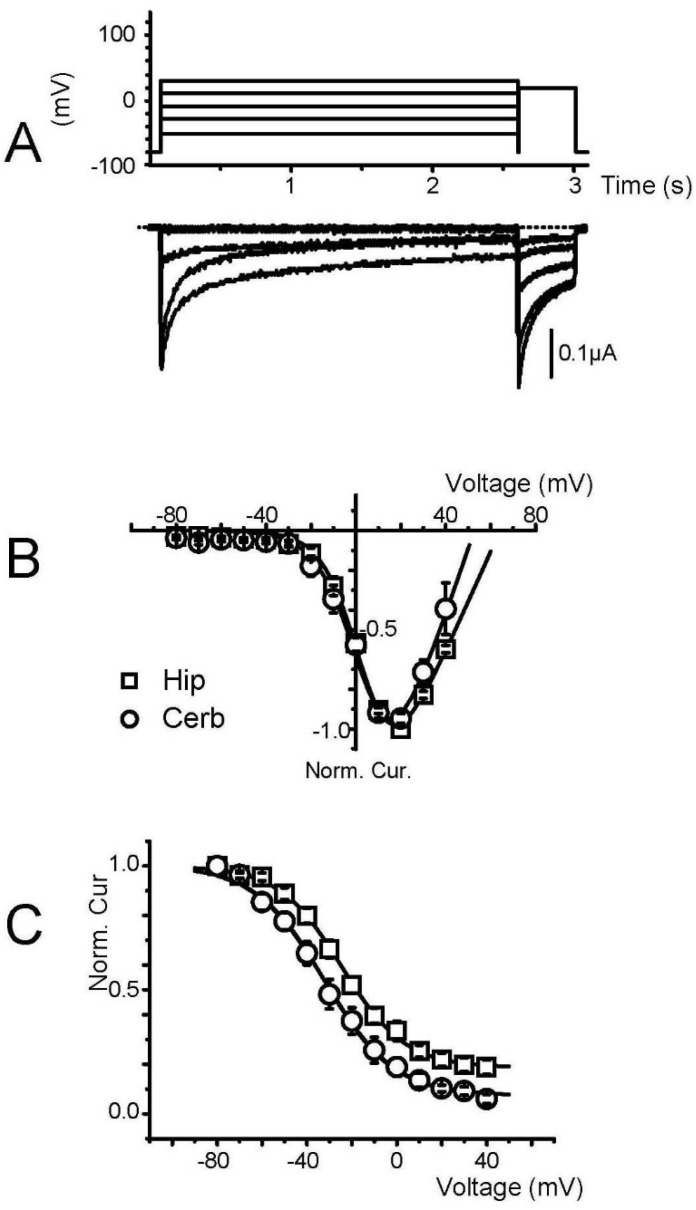
Biophysical properties of expressed voltage-gated Ca^2+^ channels. (**A**) Typical current traces recorded from an oocyte one day after injection of a hippocampal membrane preparation (7 mg/mL) during a twostep protocol. The first conditioning step had a duration of 2.5 s, and an amplitude varying from −80 to +40 mV in 10 mV increments, while the second step had a duration of 400 ms and remained at the same value ( +10 mV). Only some steps are displayed here. (**B**) Current–voltage curves obtained from similar protocols by plotting the relative peak current amplitude recorded during the first step as a function of the voltage of this step. These curves were obtained on oocytes injected with either hippocampal (Hip) or cerebellum (Cerb) membrane preparations, and Vact, kact, and Erev were 4.7 ± 0.4 mV; 7.7 ± 0.3 mV, 67 ± 1 mV (*n* = 8), and 6.8 ± 1.3 mV; and 8.4 ± 0.5 mV and 54 ± 2 mV (*n* = 11) for hippocampal and cerebellar membranes, respectively. (**C**) Isochronal inactivation curves obtained by plotting the relative peak current amplitude recorded during the second step as a function of the voltage of the first step. These curves were obtained on oocytes injected either with hippocampal (Hip) or cerebellum (Cerb) membrane preparations, and Vin, kin, and Rin were −25 ± 3 mV; 13 ± 1 mV, 0.1 ± 0.02 (*n* = 5), and −31 ± 4 mV; 15 ± 1 mV and 0.2 ± 0.01 (*n* = 10) for hippocampal and cerebellar membranes, respectively.

**Figure 4 membranes-12-00496-f004:**
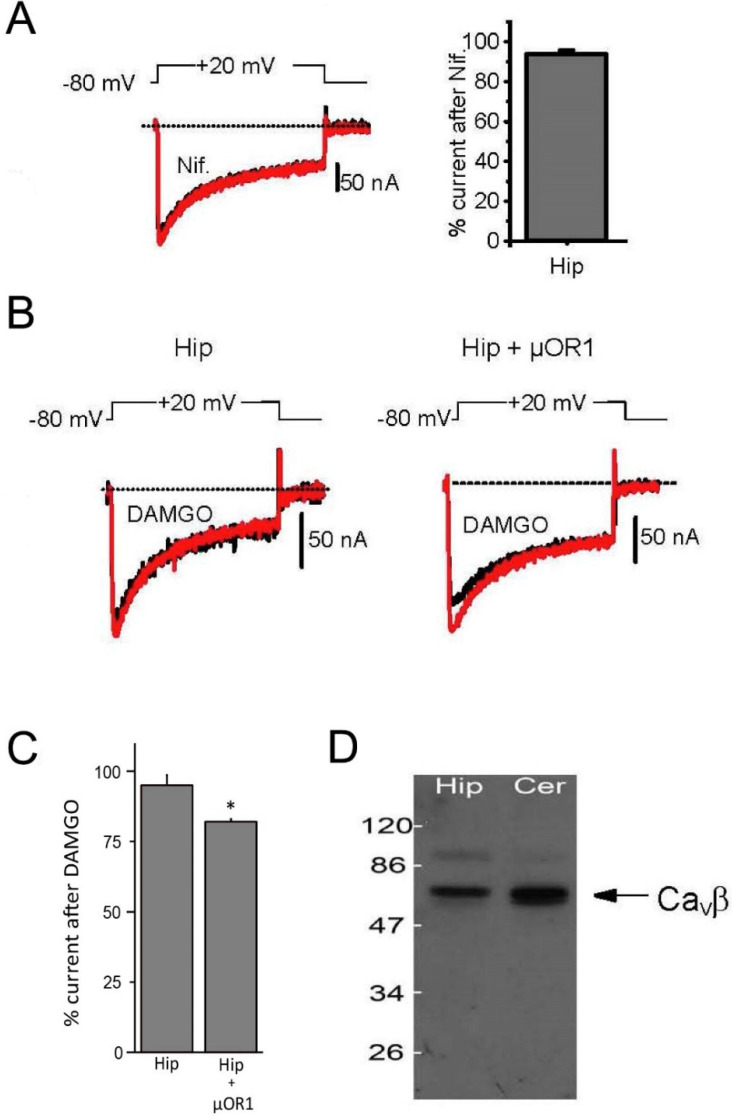
Pharmacology of the transplanted voltage-gated Ca^2+^ channels. (**A**) Typical response of a hippocampal membrane-injected oocyte to 10 µM nifedipine. Voltage-gated Ca^2+^ currents were recorded during 400 ms-long depolarizations from −80 mV to +20 mV in the presence of 40 mM Ba^2+^. Right: histogram showing the average response to nifedipine. (**B**) Response of transplanted voltage-gated Ca^2+^ channels to the µ-opioid agonist DAMGO (10 µM). Oocytes were injected with hippocampal membrane preparation, and were re-injected 24 h later with either H_2_O (Hip) or the µ1 opiod receptor RNA at 1 ug/uL (Hip + µOR1). Recordings were made 48 h after the hippocampal membrane injection. (**C**) Average response to 10 µM DAMGO of transplanted voltage-gated Ca^2+^ channels co-injected or not with the µ1 opioid receptor RNA (**D**). Western blot using either hippocampal (Hip) or cerebellum (Cer) membrane preparations and probed with an anti-Ca_V_βcom antibody showed that, in both cases, the cytoplasmic voltage-gated Ca channel auxiliary subunits, Ca_V_β, remained associated with the channel subunit(s). *—with injection.

## Data Availability

Not applicable.

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
