# Peer review of "Mammalian Brain Ca2+ Channel Activity Transplanted into Xenopus laevis Oocytes"

_membranes, 2022, doi:10.3390/membranes12050496_

Round 1

Reviewer 1 Report

This paper reports an interesting technical advance, which seems, however, too preliminary.

Specifically:

  1. Figures 1 and 2 are of secondary importance for the goal of the present study and can be moved to supplementary.
  2. Only negative results of pharmacological analysis are presented. It is utterly important to provide an efficient pharmacology in order to identify the expressed current.
  3. Many  inaccuracies (e.g. half-inactivation potential of +25 mV instead of -25 mV, "channel density" instead of "current density"). Figure captions are incomplete (e.g. Fig. 5B)
  4. English grammar requires a moderate revision, although I was able to detect multiple errors. There are more serious problems with style. 
  5. Statistical analysis is poor or absent/only representative records are given (e.g. data in Fig. 5).

Author Response

Dear Reviewer,

                First of all, we would like to thank you for your critical reading and your interesting comments and suggestions. We have addressed most of these points and revised our manuscript accordingly.

Sincerely,

The authors

Figures 1 and 2 are of secondary importance for the goal of the present study and can be moved to supplementary.

Answer: This change has been made in the manuscript: Figure 1 and table 1 have been fused to form the supplementary figure 1.

Only negative results of pharmacological analysis are presented. It is utterly important to provide an efficient pharmacology in order to identify the expressed current.

Answer: Pharmacological trials have been only made using the membrane preparation of the hippocampal fraction. We have been able to show that the nifedipine has no effect on the current meaning that the recorded current is not an L-type calcium channel. This was corroborated by the fact that the current channels recorded from the membrane preparation of the hippocampal fraction is sensitive to voltage-dependent G-protein regulation. Together these data indicate that the voltage gated calcium channels recorded are probably Cav2 type channels. Unfortunately, despite our efforts in this direction since a long time, Xenopus oocyte are not a suitable model for the pharmacology of the Cav2 channels as, until now, none of the toxins that we had tested works on Cav2 whether they come from an RNA/DNA injection or from microtransplantation.

Many  inaccuracies (e.g. half-inactivation potential of +25 mV instead of -25 mV, "channel density" instead of "current density"). Figure captions are incomplete (e.g. Fig. 5B).

Answer: These changes have been made in the manuscript.

English grammar requires a moderate revision, although I was able to detect multiple errors. There are more serious problems with style.

Answer: The manuscript have been revised by an English-native language collaborator.

Statistical analysis is poor or absent/only representative records are given (e.g. data in Fig. 5).

Answer: A 3.4 section have been added in the manuscript to specify the methods of statistical analysis.

Reviewer 2 Report

Authors studied total current of Xenopus oocyte injected with preparation of hippocampal, cerebellar and cardiac membrane. The method is in use for 30-40 years and was especially useful in the past when it was difficult to measure single channel activity. Using oocytes one is measuring hundreds if not thousand open channels of different type in the same time. Studying pharmacology of channels in oocyte preparation is difficult to interpret. At present method is obsolete and rarely used to study ion channels.

On the other hand, there is nothing wrong with the paper.

Author Response

Dear Reviewer,

                First of all, we would like to thank you for your critical reading and your interesting comments and suggestions.

One of the obvious perspectives of the microtransplantation technique is to characterize biophysical properties of ion channels (and few modulation by signaling pathways) from patients with inherited diseases. Small quantity of tissues is required. Tissues from surgery or even from autopsies could be used. It is an elegant way to bypass the ethical problems to study dysfunctions of mutated channels in their native context.

Sincerely,

The authors

Reviewer 3 Report

Manuscript of Rousset et al. entitled: “Mammalian brain Ca2+ channel activity transplanted into 2 Xenopus laevis oocytes” describes the experimental method how to prevent run-down of voltage gated Ca2+ channels and how to record activity of these channels with their native posttranslational processing, membrane environment and regulatory subunits. In my view, it is a very elegant approach how to overcome difficulties in electrophysiological recordings of voltage-gated channels in-vivo as well in-vitro. Voltage-gated Ca2+ channels are fundamental to a wide array of cellular processes such as muscle contraction and neurotransmission; therefore, a detail examination of their function related to their 3D structure (changed by numerous mutations) at the molecular level is of particular importance. Reading the manuscript and appreciating the arguments, I identified the following major and minor issues which need to be addressed by the authors. In summary, the manuscript is promising and authors should be encouraged to revise it.

Major issues:

  1. For a successful isolation of membrane fractions enriched in functional ion channels, it is essential to excise the brain and cardiac tissues as fast as possible. It is not so difficult to obtain cerebellum or cardiac ventricle; however, it is not the case for hippocampus. Could the authors mention how hippocampal tissue was exactly collected and whether the time was a critical parameter?
  2. Line 140: “ The expression of GABA receptors started …” I would not say expression, because membrane fractions were implanted. They obviously did not contain mRNA or a very low amount. 5 hours and more were highly likely needed to incorporate membrane vesicles into the plasma membrane of oocytes.
  3. As the control, water-injected oocytes were presented. A better control would be glycine-injected oocytes because pellet of membrane fractions was resuspended in glycine. Albeit, glycine was surely not responsible for Ba2+ currents.
  4. What statistics was used to compare results? It should be clearly mentioned in the Materials and Methods section.
  5. Line 222-223: The authors stated that L-type and T-type Ca2+ channels were not transplanted, even not from cardiac membrane fractions. It is quite surprising because these types of channels are the major ones in the cardiac tissue. If they were not transplanted, what Ca2+ channels generated Ba2+ currents when cardiac membrane fractions were injected (Figure 3D)? This issue should be addressed.
  6. Line 279: Western blotting just shows the presence of the given protein in a sample, but it does not demonstrate a direct interaction between proteins. Immunoprecipitation method is more appropriate to confirm a direct physical interaction between proteins.

Minor issues

  1. Figure 2: GABA should be written in all uppercase letters.
  2. In all manuscript, a space has to be added between the numerical value and unit symbol.
  3. Line 161: “40 M Ba2+” should be replaced by “40 mM Ba2+”.
  4. I recommend the authors to carefully edit the manuscript for spelling and grammar mistakes.

Author Response

Dear Reviewer,

                First of all, we would like to thank you for your critical reading and your interesting comments and suggestions. We have addressed most of these points and revised our manuscript accordingly.

Sincerely,

The authors

For a successful isolation of membrane fractions enriched in functional ion channels, it is essential to excise the brain and cardiac tissues as fast as possible. It is not so difficult to obtain cerebellum or cardiac ventricle; however, it is not the case for hippocampus. Could the authors mention how hippocampal tissue was exactly collected and whether the time was a critical parameter?

Answer: Isolation of the hippocampus is done just after cerebellum dissection and takes less than a minute. Brain is rapidly removed and placed in ice cold artificial cerebrospinal fluid and hippocampus was dissected using the same methods as the ones used to record field excitatory postsynaptic potentials.

Line 140: “The expression of GABA receptors started …” I would not say expression, because membrane fractions were implanted. They obviously did not contain mRNA or a very low amount. 5 hours and more were highly likely needed to incorporate membrane vesicles into the plasma membrane of oocytes.

Answer: This sentence was unclear and has been changed in the manuscript.

As the control, water-injected oocytes were presented. A better control would be glycine-injected oocytes because pellet of membrane fractions was resuspended in glycine. Albeit, glycine was surely not responsible for Ba2+ currents.

Answer: We have verified that 5mM Glycine have no effect on the capacity of the oocyte and do not induce any Ba2+ currents. Corresponding precisions have been added e in the manuscript.

What statistics was used to compare results? It should be clearly mentioned in the Materials and Methods section.

Answer: A 3.4 section have been added in the manuscript to specify the methods of statistical analysis.

Line 222-223: The authors stated that L-type and T-type Ca2+ channels were not transplanted, even not from cardiac membrane fractions. It is quite surprising because these types of channels are the major ones in the cardiac tissue. If they were not transplanted, what Ca2+ channels generated Ba2+ currents when cardiac membrane fractions were injected (Figure 3D)? This issue should be addressed.

We thank the reviewer for pinpointing this point. We agree that the current recorded from the cardiac preparation are probably L-type or T-type currents. Considering the biophysical properties of this current and it activation threshold (> 0 mV), it is probably L-type current. Consequently we remove from the manuscript “cerebellar and cardiac” in the section 4.3 Biophysical and pharmacological properties of expressed VGCC and limit ours conclusion about the absence of L-type calcium current to hippocampal preparation. The change have been made accordingly in the manuscript.

Line 279: Western blotting just shows the presence of the given protein in a sample, but it does not demonstrate a direct interaction between proteins. Immunoprecipitation method is more appropriate to confirm a direct physical interaction between proteins.

Answer: We agree with the reviewer that immunoprecipitation would be an adequate approach to confirm the interaction between the beta subunit and Cav subunits. However, without considering to get an efficient antibody to do such immunoprecipitation, we doubt that the quantity of material would be sufficient to detect the co-immunoprecitated subunits. However, the presence of the beta subunit in the membrane preparation combined with the biophysical properties of the calcium currents and the voltage dependent G-protein regulation of the calcium current indicate that channels responsible of these currents include a beta subunit in their macromolecular complexes.

Minor issues

  1. Figure 2: GABA should be written in all uppercase letters.

Answer: These changes have been made in the manuscript.

  1. In all manuscript, a space has to be added between the numerical value and unit symbol.

Answer: These changes have been made in the manuscript.

  1. Line 161: “40 M Ba2+” should be replaced by “40 mM Ba2+”.

Answer: These changes have been made in the manuscript.

  1. I recommend the authors to carefully edit the manuscript for spelling and grammar mistakes.

Answer: The manuscript have been revised by an English-native language collaborator.

Round 2

Reviewer 1 Report

So far, the principle issue has not been addressed properly. The authors wrote:

"Unfortunately, despite our efforts in this direction since a long time, Xenopus oocyte are not a suitable model for the pharmacology of the Cav2 channels as, until now, none of the toxins that we had tested works on Cav2 whether they come from an RNA/DNA injection or from microtransplantation."- To be honest, I do not understand the reasons for this.  If this was true (??), so how then one interpretes this: "We have been able to show that the nifedipine has no effect on the current meaning that the recorded current is not an L-type calcium channel." Because what the authors literally meant that pharmacological tools in general are futil for Xenopus oocytes, at least when they are tried against Cav channels. Anyway, this note requires much more justification, perhaps, recrouting and discussing the data by others on this point. I can not accept the arguments by authors in the present form.   

Reviewer 3 Report

The most of my concerns were appropriately addressed. But I have one more suggestion regarding Fig. 4. Bar graphs showing and comparing effects of DAMGO on the Ba2+ currents in hippocampal membrane-injected oocytes and when the µOR1 receptor was expressed should be implemented. It should be clear to readers that DAMGO did have a significant inhibitory effect on the Ba2+ current when expression of the µOR1 receptor was induced.  

Author Response

Dear reviewer 3,

We would like to thank you again for your critical reading, and your comment. 

Sincerely,

The authors

The most of my concerns were appropriately addressed. But I have one more suggestion regarding Fig. 4. Bar graphs showing and comparing effects of DAMGO on the Ba2+ currents in hippocampal membrane-injected oocytes and when the µOR1 receptor was expressed should be implemented. It should be clear to readers that DAMGO did have a significant inhibitory effect on the Ba2+ current when expression of the µOR1 receptor was induced.

Answer: We agree with the reviewer 3. This change has been made in the manuscript: Figure 4C now present such bar graph.

Round 3

Reviewer 1 Report

N/A